# Federated Fuzzy C-Means with Schatten $p$-Norm Minimization

## ABSTRACT

Federated multi-view clustering aims to provide a feasible and effective solution for handling unlabeled data owned by multiple clients. There are two main challenges: 1) The local data is always sensitive, thus preventing any inadvertent data leakage to the server or other clients. 2) Multi-view data contain both consistency and complementarity information, necessitating thorough exploration and utilization of these aspects to achieve enhanced clustering performance. Fully considering the above challenges, in this paper, we propose a novel federated multi-view method named **F**ederated **F**uzzy **C-M**eans with **S**chatten-**p** Norm Minimization(FFCMSP) which is based on Fuzzy C-Means and Schatten $p$-norm. Specifically, we utilize the membership degrees to replace conventional hard clustering results in K-means, enabling improved uncertainty handling and less information loss. Moreover, we introduce a Schatten $p$-norm-based regularizer to fully explore the inter-view complementary information and global spatial structure. Correspondingly, we also proposed a federated optimization algorithm enabling clients to collaboratively learn the clustering results. Extensive experiments on several datasets demonstrate that our proposed method exhibits superior performance in federated multi-view clustering.

## CCS CONCEPTS

• **Computing methodologies** → **Machine learning**; *Artificial intelligence*.

## KEYWORDS

Fuzzy C-Means, Federated, multi-view clustering, tensor Schatten $p$-norm

## 1 INTRODUCTION

Multi-view data refers to data containing information from multiple perspectives (*e.g.*, modalities, sources, and viewpoints), and each view represents a distinct observation of the same object, thus, they generally have consistent and complementary information, which is beneficial for several applications. For example, by leveraging diverse types of medical data, such as CT scans and electrocardiograms, hospitals can obtain a more comprehensive assessment of a patient's physical condition. However, multi-view data learning confronts several challenges because the high cost of annotating complex multi-view data makes it difficult to acquire a substantial

**Unpublished working draft. Not for distribution.**

number of high-quality labels. Thus, multi-view clustering has attracted considerable attention as an unsupervised learning approach [5, 35].

Existing multi-view clustering methods can be summarised into two categories [12]: neural network-based multi-view clustering and heuristic-based multi-view clustering. Furthermore, neural network-based multi-view clustering can be categorized into deep representation learning [40, 43] and deep graph learning [44, 45] while heuristic-based multi-view clustering can be categorized into non-negative matrix factorization [19, 30], graph learning [20, 28], latent representation learning [22, 46], and tensor learning [33].

Even though existing multi-view clustering methods achieve promising performance, they cannot be applied to scenarios where multi-view data is distributed across multiple clients, while clients are unwilling to share their data for the sake of privacy. Inspired by federated learning, which enables multiple clients to cooperatively train a global model without revealing any sensitive information [34], federated multi-view learning is developed. For example, Huang et al. extended multi-view clustering based on non-negative matrix factorization (NMF) to construct a federated multi-view clustering method named FedMVL [19] by dividing the global optimization problem into multiple sub-problems that can be solved locally to fully protect local data privacy.

However, FedMVL cannot obtain optimal performance because the underlying K-means enforces the clustering result to assign a concrete cluster label to each label and thus suffers from sensitivity to outliers and . Besides, it separates feature extraction and clustering into two steps and results in performance degradation. For another, FedMVL first learns each view's feature via NMF separately and then fuses multi-view features for clustering. Thus, it is unable to exploit the inter-view spatial structure of multi-view data.

To overcome these weaknesses, we propose a novel **F**ederated **F**uzzy **C-M**eans with **S**chatten-$p$ Norm Minimization (FFCMSP). Compared with FedMVL, FFCMSP replaces K-means with fuzzy C-Means. Since fuzzy C-Means directly learn soft assignment via membership degree matrix, it alleviates the sensitivity to outliers due to the hard assignment of K-means and caused by two-step clustering. Besides, inspired tenser-based methods [33, 48], we construct a tensor from the membership degree matrix of each view and validate Schatten $p$-norm based loss on it to learn the inter-view spatial structure. We further develop a distributed optimization algorithm, which enables all clients to optimize the model locally and collaboratively. Specifically, our contributions are as follows:

- We introduce fuzzy C-Means to federated multi-view clustering to overcome the weaknesses of K-means. It utilizes the membership degree instead of hard assignment to capture the overall structure of distributed data with heterogenic features.
- We construct a membership degree tensor from each view's membership degree matrix, and by minimizing the Schatten $p$-norm based regularizer, it can better explore the relationship between views and the complementary information of different views.

- To enable clients to cooperatively train a global model without any privacy data leakage, we developed a federated optimization algorithm to solve the tensor-based multi-view clustering objective.
- We conducted extensive experiments on multiple datasets to evaluate the model performance and compare it with several state-of-the-art methods, whose results demonstrate the superiority of our proposed approach.

## 2 RELATED WORK

### 2.1 Multi-View Clustering

Multi-view clustering aims to partition multi-view data into different clusters, which is widely applied in multi-view data labeling and pre-processing. Existing multi-view clustering methods include neural network-based multi-view clustering and heuristic multi-view clustering [12]. Although neural network based methods exhibit outstanding performance due to their ability to extract deep and nonlinear features of multi-view, their training cost is extremely high and is not suitable for lightweight applications. Differently, heuristic has lower computational complexity and better interpretability, which has attracted more attention recently.

There are mainly four categories of heuristic multi-view clustering methods: non-negative matrix factorization-based (NMF) methods [19, 30], graph learning-based methods [20, 28], latent representation learning methods [22, 46], and tensor learning-based methods [33]. The main idea of NMF-based methods is to factorize the data matrix of each view into a coefficient matrix and a basis matrix with lower dimensions for clustering. For example, Liu et al. [30] proposed joint multi-view NMF to learn a common consensus coefficient matrix; Liang et al. [29] introduced co-orthogonal constraints to capture the diversity within views and learn the orthogonal basis matrices. Multi-view graph clustering attempts to learn a consistent clustering structure from the correlation of samples, which is composed of graph fusion and graph partition. Wang et al. [41]firstly perceived the correlations between samples under the same view, and then all views adaptively collaborated to construct the consensus graph. Latent representation learning-based methods learn the shared latent representation of each view for clustering. [24] presents regularized and hybrid multiview coding (RHMC), which employs self-supervised learning to enhance the discriminative information of the shared feature.

The aforementioned methods all process multi-view data separately and then fuse the learned features or clustering results. To better exploit the inter-view spatial structure, tensor-based methods compose multi-view data into a three-order tensor. For instance, Lu et al. [33] constructed the third-order tensor from view-specific label matrices and minimized the divergence between the matrices to fully explore the complementary information between views. Li et al. [23] extended NMF into orthogonal non-negative tensor factorization and introduced tensor Schatten $p$-norm regularizer to fully utilize the complementary information provided by each view. Nevertheless, these methods are designed for centralized settings and ignore the possibility that multi-view data are held by different entities.

### 2.2 Federated Learning

Federated learning [34] is a distributed learning paradigm that enables multiple entities to collaboratively train a global model on data distributed across multiple devices without revealing any private information. The concept of federated learning and its specific algorithm, FedAVG, were proposed simultaneously [34]. FedAVG involves performing stochastic gradient descent on each client while aggregating the local trained models on the server by performing averaging. Afterward, many variants of federated learning were proposed: FedAMP [21], FedProx [27]. Federated learning can be categorized into three types: Horizontal Federated learning (HFL), vertical federated learning(VFL), and federated transfer learning (FTL).

In HFL, several clients share the same feature spaces but different sample spaces. [14] introduced HFL for Electroencephalography classification techniques to meet the requirements of privacy protection and data heterogeneity. Moreover, [47] focused on measuring the contribution to the model of all clients and the global server evaluates the contribution through reinforcement learning techniques. For VFL, several clients share the same sample space but different feature spaces. [6] proposed a novel method that enables each client to execute stochastic gradient algorithms independently, and a new technique of perturbed local embedding is proposed. Traditional VFL identifies the shared sample by Private Set Intersection(PSI), which may lead to some data leakage. Thus, [39] proposed a Private Set Union(PSU) based VFL framework to avoid such kind of data leakage. For FTL, clients share both different sample spaces and feature spaces. [31] first proposed the concept of FTL, enabling a target-domain party to build flexible and effective models by leveraging rich labels from a source domain. All the previously mentioned methods are designed for single-view data and fail to fully exploit the latent information inherent in multi-view data for learning.

### 2.3 Federated Multi-view Clustering

Traditional multi-view clustering methods generally assume that all data are held by a single party and never consider the application in which multi-view data are distributed across different clients and privacy-preservation. To tackle this problem, federated multi-view learning was proposed, and there are several federated multi-view learning methods developed for different settings. Multi-view learning was introduced to the personalized recommendation system [10], cancer subtype identification [7], and other domains.

However, similarly, labeling multi-view data is costly, several times more expensive than labeling single-view data. This led to the emergence of federated multi-view clustering(FMVC). [4] introduced FMVC to the medical field. The proposed method applies to both VFL and HFL and keeps sensitive data private. [8] is a federated multi-view clustering method based on neural networks and guarantees data security by transmitting non-sensitive parameters between clients and the global server. Neural network-based FMVC methods are always time-consuming. Thus, most FMVC methods are based on heuristic clustering methods. [19] combines NMF and K-means to construct a time-efficient FMVC framework and the final clustering result comes from the coefficient matrix. However, because of using K-means on the coefficient matrix, we cannot effectively capture the relationship between samples and clusters, leading

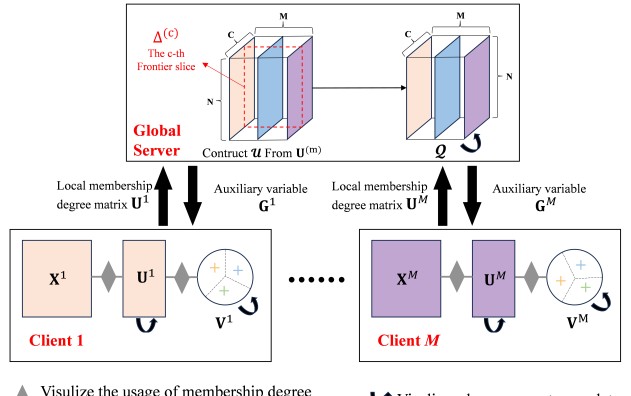

**Figure 1: The framework of the proposed FFCMSP. The federated framework contains one global server and $M$ local clients. $\mathbf{X}^m$, $\mathbf{V}^m$ denotes the data matrix and center cluster centroid matrix. The grey ◆ represents that the membership matrix U is used to describe the membership relationship between samples and cluster centroid. Moreover, black ↻ visually represents whether the parameters are updated on the client or the server. The contruction of tenser $\mathcal{U}$ is also visualized. Finally, the $\Delta^{(c)}$ denotes the $c$-th frontier slice of $\mathcal{U}$ and it well characterize the complementary information embedded in inter-views. [15]**

to some information loss and performance degradation. Therefore, [17] proposed fuzzy C-Means-based FMVC which uses membership degree instead of hard clustering assignment. Nevertheless, it fails to capture the consistent information between views.

## 3 METHOD

### 3.1 Preliminaries

For easier presentation, we introduce Schatten $p$-norm [32] as follows.

DEFINITION 1. *The matrix Schatten p-norm for matrix* $\mathbf{M} \in \mathbb{R}^{n_1 \times n_2}$ *is defined as below:*

$$||\mathbf{M}||_{sp} = \left( \sum_{i=1}^{\min(n_1, n_2)} \sigma_i^p \left( \mathbf{M} \right) \right)^{\frac{1}{p}} \tag{1}$$

*where* $\sigma_i \left( \mathbf{M} \right)$ *is the i-th singular value of matrix* $\mathbf{M}$.

DEFINITION 2 (TENSOR SINGULAR VALUE DECOMPOSITION, T-SVD). *Given a tensor* $\mathcal{M} \in \mathbb{R}^{n_1 \times n_2 \times n_3}$, *then we have its t-SVD:*

$$\mathcal{M} = \mathcal{U} * \mathcal{S} * \mathcal{V}^T \tag{2}$$

*where* $\mathcal{U} \in \mathbb{R}^{n_1 \times n_1 \times n_3}$, $\mathcal{V} \in \mathbb{R}^{n_2 \times n_2 \times n_3}$ *are both orthogonal tensors, and* $\mathcal{S} \in \mathbb{R}^{n_1 \times n_2 \times n_3}$ *is diagonal tensor. Moreover, all frontal slices of* $\mathcal{S}$ *are diagonal matrixes, and the i-th singular value of k-th frontal slices of* $\mathcal{S}$ *is denoted as* $\mathcal{S}^k \left( i \right)$.

DEFINITION 3 (TENSOR SCHATTEN $p$-NORM). *Based on **Definition** 1 and **Definition** 2, the tensor Schatten p-norm for tensor* $\mathcal{M}$

*is defined as below:*

$$||\mathcal{M}||_{s_p} = \left( \sum_{i=1}^{\min(n_1, n_2)} \sum_{k=1}^{n_3} \left( \mathcal{S}^k \left( i \right) \right)^p \right)^{\frac{1}{p}} \tag{3}$$

where $0 \leq p \leq 1$, tensor Schatten $p$-norm has proven to be a better rank approximation method than the tensor nuclear norm (TNN), which motivates us to apply it in our work to learn more robust and clustering results. Given a typical Tensor Schatten $p$-norm problem, it can be solved with the following Lamma.

LAMMA 1. *For* $\mathcal{B} \in \mathbb{R}^{n_1 \times n_2 \times n_3}$ *and* $\mathcal{C} \in \mathbb{R}^{n_1 \times n_2 \times n_3}$, *the function*

$$\min_{\mathcal{B}} \tau ||\mathcal{B}||_{s_p}^p + \frac{1}{2} ||\mathcal{B} - \mathcal{C}||_F^2 \tag{4}$$

*has the optimal solution given by*

$$\mathcal{B}^* = \text{ifft} \left( \mathcal{U} * \mathcal{D}_{\tau, p} \left( \mathcal{C} \right) * \mathcal{V}^T \right) \tag{5}$$

*where* $\mathcal{U}$ *and* $\mathcal{V}$ *are obtained via t-SVD of* $\mathcal{C}$, *i.e.* $\mathcal{C} = \mathcal{U} * \mathcal{S} * \mathcal{V}^T$, *and* $\mathcal{D}_{\tau, p} \left( \mathbf{C}^{(m)} \right) = \text{diag} \left( \zeta \left( \mathbf{C}^{(m)} \right) \right)$, $\zeta \left( \mathbf{C}^{(m)} \right) = \text{GST} \left( \sigma \left( \mathbf{S}^{(m)}, \tau, p \right) \right)$. *The GST algorithm is introduced in [15].*

### 3.2 Objective Function

**Problem Statement:** We consider the same federated multi-view settings as that in [19], which includes a centralized server $\mathcal{S}$ and $M$ clients. Multi-view data, denoted as $\mathbf{X} = \{ \mathbf{X}^{(1)}, \mathbf{X}^{(2)}, \cdots \mathbf{X}^{(M)} \}$, is held by different clients. Specifically, the $m$-th view $\mathbf{X}^{(m)} \in \mathbb{R}^{N \times d^{(m)}}$ ($m = 1, 2, \cdots, M$) is distributed in the $m$-th client, $C_m$, where $d^{(m)}$ is the feature dimension, $N$ is sample number, and $M$ is the number of views as well as client number. Our goal is enable the server and clients to cooperatively train a global clustering model without exchanging any raw data.

**Federated Multi-View Fuzzy C-Means:** Our work is derived from fuzzy C-Means (FCM) [1], which assigns data points to clusters with membership degree matrix. Suppose there are $N$ samples of dimension $D$ denoted by $\mathbf{X} \in \mathbb{R}^{N \times D}$, FCM divides them into $C$ clusters with the following objective:

$$\min_{U, V} \sum_{i=1}^{N} \sum_{j=1}^{C} u_{ij}^t ||\mathbf{x}_i - \mathbf{v}_j||_F^2$$

$$s.t. \quad \sum_{j=1}^{C} u_{ij} \geq 0 = 1, u_{ij} \geq 0, t \geq 1 \tag{6}$$

where $u_{ij}$ is the membership degree, representing the probability that the $i$-th sample belongs to the $j$-th cluster; $\mathbf{V} \in \mathbb{R}^{C \times D}$ is the clustering centroids matrix; $t$ is the fuzzification coefficient, and bigger $t$ tends to defocus membership towards the fuzziest state [1], whose suggested range is $1 \leq t \leq 3$.

Compared with K-means, FCM can better resist noises or outliers, motivated by which, we adopt multi-view FCM as our underlying multi-view clustering method, which can be expressed as follows:

$$\min_{U^{(m)}, V^{(m)}} \sum_{m=1}^{M} \sum_{i=1}^{N} \sum_{j=1}^{C} \left( u_{ij}^{(m)} \right)^t ||\mathbf{x}_i^{(m)} - \mathbf{v}_j^{(m)}||_F^2$$

$$s.t. \quad \sum_{j=1}^{C} u_{ij}^{(m)} = 1, u_{ij}^{(m)} \geq 0, t \geq 1 \tag{7}$$

where $\mathbf{X}^{(m)}$ is the data matrix located on the $m$-th clients and $\mathbf{V}^{(\mathbf{m})}$ is the corresponding cluster centroids. $t$ is also the fuzzification coefficient, similarly. To simplify the optimization process, this paper sets the fuzzification coefficient $t$ to 1. Thus, based on **Eq.** (7), the objective function is:

$$\min_{U^{(m)},V^{(m)}} \sum_{m=1}^{M} \sum_{i=1}^{N} \sum_{j=1}^{C} u_{ij}^{(m)} ||\mathbf{x}_i^{(m)} - \mathbf{v}_j^{(m)}||_F^2$$

$$s.t. \quad \sum_{j=1}^{C} u_{ij}^{(m)} = 1, u_{ij}^{(m)} \geq 0 \tag{8}$$

**Tensor Schatten $p$-Norm Based Regularizer**: By observing **Eq.** (8), we find that in each view, each sample corresponds to a membership degree $u_{ij}$ for each cluster, and for $N$ samples and $C$ clusters, the membership degrees form a matrix $\mathbf{U}^{(m)} \in \mathbb{R}^{N \times C}$, thus latent information is hidden between different view-specific membership degree matrix. Inspired by [33], we introduce a regularizer based on the tensor Schatten $p$-norm to enforce the model to learn the global spatial structure and inter-view complementary information, the final objective function becomes:

$$\min_{U^{(m)},V^{(m)}} \sum_{m=1}^{M} \sum_{i=1}^{N} \sum_{j=1}^{C} u_{ij}^{(m)} ||\mathbf{x}_i^{(m)} - \mathbf{v}_j^{(m)}||_F^2 + \lambda ||\mathcal{U}||_{sp}^p$$

$$s.t. \quad \sum_{j=1}^{C} u_{ij}^{(m)} = 1, u_{ij}^{(m)} \geq 0 \tag{9}$$

where $\lambda$ is the trade-off parameter and the membership degree tensor $\mathcal{U}$ is constructed by stacking all views' local membership degree matrix and taking a simple rotation as shown in **Fig.** 1.

## 4 OPTIMIZATION ALGORITHM

In centralized scenarios, **Eq.** (9) can be easily solved. However, with the multi-view data distributed to local clients, our design goal is to employ federated learning to collaboratively learn a clustering model while considering the requirement of data privacy. This requires that the raw models or parameters transmitted between clients and the global server should be non-sensitive, which poses extra challenges of optimization. To address this issue, we further develop a federated optimization algorithm in this scenario.

Firstly, we introduced augmented Lagrange multiplier(ALM) to **Eq.** (9):

$$\min \mathcal{L}(\mathcal{U}, \mathcal{G}, \mathcal{V}) = \sum_{m=1}^{M} \sum_{i=1}^{N} \sum_{j=1}^{C} u_{ij}^{(m)} ||\mathbf{x}_i^{(m)} - \mathbf{v}_j^{(m)}||_F^2$$

$$+ \lambda ||\mathcal{G}||_{sp}^p + \langle \mathcal{Q}, \mathcal{U} - \mathcal{G} \rangle + \frac{\mu}{2} ||\mathcal{U} - \mathcal{G}||_F^2 \tag{10}$$

$$s.t. \quad \sum_{j=1}^{C} u_{ij}^{(m)} = 1, u_{ij}^{(m)} \geq 0$$

where $\mu$ is the penalty parameter, $\mathcal{Q}$ is the Lagrange multiplier, and $\langle \cdot \rangle$ denotes the inner product operation. The optimization algorithm is divided into three parts, as described below.

**Solving $U^{(m)}$ with fixed $\mathcal{G}$ and $\mathcal{V}$:** with fixed $\mathcal{G}$ and $\mathcal{V}$, the objective function becomes:

$$\min \sum_{m=1}^{M} \sum_{i=1}^{N} \sum_{j=1}^{C} u_{ij}^{(m)} ||\mathbf{x}_i^{(m)} - \mathbf{v}_j^{(m)}||_F^2 + \langle \mathcal{Q}, \mathcal{U} - \mathcal{G} \rangle + \frac{\mu}{2} ||\mathcal{U} - \mathcal{G}||_F^2$$

$$s.t. \quad \sum_{j=1}^{C} u_{ij}^{(m)} = 1, u_{ij}^{(m)} \geq 0 \tag{11}$$

Considering the independence of $\mathcal{U}$, $\mathcal{Q}$, and $\mathcal{G}$, **Eq.** (10) could be decomposed into $M$ independent subproblems, where the $m$-th subproblem is:

$$\min \sum_{i=1}^{N} \sum_{j=1}^{C} u_{ij}^{(m)} ||\mathbf{x}_i^{(m)} - \mathbf{v}_j^{(m)}||_F^2 + \left\langle \mathbf{Q}^{(m)}, \mathbf{U}^{(m)} - \mathbf{G}^{(m)} \right\rangle$$

$$+ \frac{\mu}{2} ||\mathbf{U}^{(m)} - \mathbf{G}^{(m)}||_F^2 \tag{12}$$

$$s.t. \quad \sum_{j=1}^{C} u_{ij}^{(m)} = 1, u_{ij}^{(m)} \geq 0$$

Through simple algebra, **Eq.** (12) can be transformed into:

$$\min \sum_{i=1}^{N} \sum_{j=1}^{C} u_{ij}^{(m)} \mathbf{D}_{ij}^{(m)} + \frac{\mu}{2} ||\mathbf{U}^{(m)} - \mathbf{T}^{(m)}||_F^2$$

$$s.t. \quad \sum_{j=1}^{C} u_{ij}^{(m)} = 1, u_{ij}^{(m)} \geq 0 \tag{13}$$

where $\mathbf{D}_{ij}^{(m)} = ||\mathbf{x}_i^{(m)} - \mathbf{v}_j^{(m)}||_F^2$, and $\mathbf{T}^{(m)} = \mathbf{G}^{(m)} - \frac{\mathbf{Q}^{(m)}}{\mu}$. By introducing the Karush-Kuhn-Tucker (KKT) condition, we have:

$$u_{ij}^{(m)} = \frac{C\mathbf{T}_{ij}^{(m)} - \sum_{j=1}^{C} \mathbf{T}_{ij}^{(m)}}{C} + \frac{C\mathbf{D}_{ij}^{(m)} - \sum_{j=1}^{C} \mathbf{D}_{ij}^{(m)}}{C\mu} + \frac{1}{C} \tag{14}$$

**Solving $\mathcal{G}$ with fixed $\mathcal{U}$ and $\mathcal{V}$:** With fixed $\mathcal{U}$ and $\mathcal{V}$, the objective function concerning $\mathcal{G}$ becomes:

$$\min \lambda ||\mathcal{G}||_{sp}^p + \langle \mathcal{Q}, \mathcal{U} - \mathcal{G} \rangle + \frac{\mu}{2} ||\mathcal{U} - \mathcal{G}||_F^2$$

$$= \min \lambda ||\mathcal{G}||_{sp}^p + \frac{\mu}{2} ||\mathcal{G} - \mathcal{P}||_F^2 \tag{15}$$

where $\mathcal{P} = \mathcal{U} - \frac{\mathcal{Q}}{\mu}$. From **Lamma.** 1, we get the optimal solution of **Eq.** (15):

$$\mathcal{G}^* = \text{ifft} \left( \mathcal{U} * \mathcal{D}_{\frac{\lambda}{\mu}, p} (\mathcal{P}) * \mathcal{V}^T \right) \tag{16}$$

where $\mathcal{U}$ and $\mathcal{V}_*$ are obtained via t-SVD of $\mathcal{P}$, *i.e.*, $\mathcal{P} = \mathcal{U} * \mathcal{S} * \mathcal{V}_*^T$.

**Solving $\mathbf{V}^{(m)}$ with fixed $\mathcal{U}$ and $\mathcal{G}$:** With fixed $\mathcal{U}$ and $\mathcal{V}$, the objective function concerning $\mathcal{V}$ becomes:

$$\min \sum_{m=1}^{M} \sum_{i=1}^{N} \sum_{j=1}^{C} u_{ij}^{(m)} ||\mathbf{x}_i^{(m)} - \mathbf{v}_j^{(m)}||_F^2 \tag{17}$$

Similarly, we decompose **Eq.** (17) into $M$ independent subproblems owing to the independence of $\mathcal{U}$, and the $m$-th subproblem is defined as:

$$\min \mathcal{L}_{\mathbf{V}^{(m)}} = \min \sum_{i=1}^{N} \sum_{j=1}^{C} u_{ij}^{(m)} ||\mathbf{x}_i^{(m)} - \mathbf{v}_j^{(m)}||_F^2 \tag{18}$$

We take the partial derivative of **Eq.** (18) concerning $\mathbf{v}_j^{(m)}$ and set it to zero:

$$\sum_{i=1}^{N} u_{ij}^{(m)} \mathbf{x}_i^{(m)} = \sum_{i=1}^{N} u_{ij}^{(m)} \mathbf{v}_j^{(m)}$$

$$\mathbf{v}_j^{(m)} = \frac{\sum_{i=1}^{N} u_{ij}^{(m)} \mathbf{x}_i^{(m)}}{\sum_{i=1}^{N} u_{ij}^{(m)}} \tag{19}$$

### 4.1 Federated workflow

Considering local client data security, the data or parameters transmitted between the client and the server must be insensitive. Therefore, based on the above optimization algorithm, we propose the workflow in the federated setting. By observing **Eq.** (14), **Eq.** (16), and **Eq.** (19), we draw the following conclusions:

- The update of $u_{ij}^{(m)}$ is related to $d_{ij}^{(m)}$, $\mathbf{G}^{(m)}$, and $\mathbf{Q}^{(m)}$, where $d_{ij}^{(m)}$ describes the is between $i$-th sample and $j$-th cluster centroid and it can be calculated locally. Therefore, each client can update $u_{ij}^{(m)}$ locally as long as they have a backup of $\mathbf{G}^{(m)}$ and $\mathbf{Q}^{(m)}$.
- The update of $\mathcal{G}$ is related to $\mathcal{U}$ and $\mathcal{Q}$, thus, it can only be performed globally and $\mathbf{U}^{(m)}$ should be transmitted to the global server in advance.
- The update of $\mathbf{V}^{(m)}$ is related to $\mathbf{U}^{(m)}$ and $\mathbf{X}^{(m)}$, and it can be performed locally because the relevant data is stored locally.
- The optimal solution of $\mathcal{Q}$ is $\mathcal{Q} \leftarrow \mathcal{Q} + \mu(\mathcal{U} - \mathcal{G})$, which can only be performed globally because it requires tensor $\mathcal{U}$.

After the above analysis, we described the detailed workflow in the federated setting.

(1) Firstly, each client $C_m$ initializes $u_{ij}^{(m)} = \frac{1}{C}$ and $\mathbf{V}^{(m)}$ randomly, and the global server initializes $\mathcal{G} = \mathcal{U}$ and $\mathcal{Q} = \mathbf{0}$.

(2) In each communication round, each client $C_m$ updates $\mathbf{U}^{(m)}$ with **Eq.** (14) and $\mathbf{V}^{(m)}$ with **Eq.** (19) locally and then sends updated $\mathbf{U}^{(m)}$ to the global server $\mathcal{S}$.

(3) After receiving them, $\mathcal{S}$ stacks all $\mathbf{U}^{(m)}$ into a tensor $\mathcal{U}$, and then update $\mathcal{G}$ with **Eq.** (16) and $\mathcal{Q}$ with $\mathcal{Q} \leftarrow \mathcal{Q} + \mu(\mathcal{U} - \mathcal{G})$. The global server then transmits the $m$-th slice of updated $\mathcal{G}$ and $\mathcal{G}$, $\mathbf{G}^{(m)}$ and $\mathbf{Q}^{(m)}$, to the corresponding client.

(4) The step (2) and (3) repeat until convergence, and then, the final cluster assignment matrix $\mathbf{H}$ is calculated globally by:

$$h_{ij} = \begin{cases} 1 & j = \arg\min_p \left( \sum_{m=1}^{M} \mathbf{u}_i^{(m)} \right)_p \\ 0 & \text{otherwise} \end{cases} \tag{20}$$

The detailed workflow is summarized in *Algorithm* 1.

### 4.2 Complexity Analysis

*4.2.1 Computational Complexity.* The complexity of FFCSP should be analyzed from two aspects, i.e., local client-side and global server-side.

**Local client-side:** The computation load on each client includes the update of $\mathbf{U}^{(m)}$ and $\mathbf{V}^{(m)}$. In each local iteration, the computational complexity of the update of $\mathbf{U}^{(m)}$ is $O\left(NC\left(d^{(m)} + C\right)\right)$ while the computational complexity of the update of $\mathbf{V}^{(m)}$ is $O(NC)$. Thus,

---

**Algorithm 1** Federated Fuzzy C-Means with Schatten $p$-Norm Minimization(FFCMSP)

**input:** The data $\mathbf{X} = \{\mathbf{X}^{(1)}, \mathbf{X}^{(2)}, ..., \mathbf{X}^{(M)}\}$ on $M$ local clients, cluster number $C$; parameters: $\mu$, $p$, $\lambda$;
**output:** Output: Cluster assignment $\mathbf{H}$;
1: Client initialization: Initialize $u_{ij}^{(m)} = \frac{1}{C}$ and $\mathbf{V}^{(m)}$ randomly.
2: Server initialization: Initialize $\mathcal{Q} = \mathcal{U}$ and $\mathcal{G}$ to be all zero.
3: **while** not converged **do**
4:     **for** $m = 1$ $to$ $M$ **do**
5:                           ▷ **On $m$-th client $C_m$**
6:         Update membership matrix $\mathbf{U}^{(m)}$ according to (14);
7:         Update cluster centroids matrix $\mathbf{V}^{(m)}$ according to (19);
8:         Send updated $\mathbf{U}^{(m)}$ to the global server;
9:     **end for**
10:                         ▷ **On Server $\mathcal{S}$**
11:     Stack all $\mathbf{U}^{(m)}$ to construct tensor $\mathcal{U}$ as shown in **Fig.** 1;
12:     Update $\mathcal{G}$ with new $\mathcal{U}$ according to (16);
13:     Update $\mathcal{Q}$ according to $\mathcal{Q} = \mathcal{Q} + \mu(\mathcal{U} - \mathcal{G})$;
14:     Send $\mathbf{G}^{(m)}$ and $\mathbf{Q}^{(m)}$ to $C_m$;
15: **end while**
16: $\mathcal{S}$ aggregates $\mathbf{U}^{(m)}$ into cluster assignment matrix $\mathbf{H}$ according to (20);
17: **return H**

---

the overall computational complexity of updating local model $ITER_L$ iterations is $O\left(ITER_L NC\left(d^{(m)} + C\right)\right)$.

**Global client-side:** The computational load on the global server includes the update of $\mathcal{G}$ and $\mathcal{Q}$. In each global iteration, the computational complexity of the update of $\mathcal{G}$ is $O\left(2NCM\log(CM) + NCM^2\right)$, while the computational complexity of the update of $\mathcal{Q}$ is $O(NCM)$. Thus, the overall computational complexity of updated global model $ITER_G$ iterations is $O\left(ITER_G\left(2NCM\log(CM) + NCM^2\right)\right)$.

*4.2.2 Communication Complexity.* Moreover, we analyze the transmission load between each client and the global server.

**Local Client→Global Server:** Each client transmits its local $\mathbf{U}^{(m)}$ to the global server before aggregating. For all clients, the amount of data transmitted in each communication round is $O(NCM)$.

**Global Server→Local Client:** The global server sends the $m$-th slice of $\mathcal{G}$ and $\mathcal{Q}$ to the m-th client after aggregating. Thus, the total amount of data transmitted in each communication round is $O(2NCM)$.

## 5 EXPERIMENT

### 5.1 Experimental Settings

We validate our model on eight multi-view datasets and compare it with several state-of-the-art methods. We implement all the methods with MATLAB2023b on a Desktop running Windows 11 equipped with Intel Core i5-13400CPU(2.50GHz) and 32GB DDR4 RAM. For the federated setting, our experiment consists of the same number of clients as the number of views and one server, while each client stores the data with one view. We use three widely used metrics, i.e., Accuracy (ACC), Normalized Mutual Information(NMI), and Purity(PUR) to evaluate our method.

| Datasets | 3-sources | | | BBCSport | | | Sonar | | | Caltech-5v | | |
|---|---|---|---|---|---|---|---|---|---|---|---|---|
| Metrics | ACC | NMI | PUR | ACC | NMI | PUR | ACC | NMI | PUR | ACC | NMI | PUR |
| DiMSC | 70.81 | 63.81 | 76.13 | 82.17 | 64.07 | 82.17 | 56.41 | 1.63 | 56.41 | 57.57 | 39.76 | 61.43 |
| MvLRSSC | 54.67 | 44.92 | 63.31 | 64.07 | 40.92 | 65.07 | 50.48 | 0.01 | 53.37 | 46.15 | 34.81 | 46.79 |
| RMSL | 34.91 | 14.43 | 42.60 | 76.63 | 72.36 | 76.63 | 50.48 | 1.76 | 53.37 | 55.00 | 52.18 | 59.07 |
| GMC | 69.23 | 62.16 | 74.56 | 80.70 | 76.00 | 79.43 | 50.48 | 4.50 | 53.37 | 34.07 | 48.40 | 36.07 |
| MvDGNMF | 66.27 | 48.77 | 70.41 | 85.11 | 70.07 | 85.11 | 63.94 | 6.00 | 63.94 | 49.57 | 38.24 | 53.86 |
| UDBGL | 34.91 | 5.60 | 35.50 | 36.40 | 2.43 | 36.58 | 57.21 | 1.61 | 57.21 | 31.80 | 23.54 | 19.28 |
| FastMICE | 52.07 | 45.57 | 66.27 | 50.00 | 25.49 | 55.33 | 58.65 | 3.63 | 58.65 | 77.58 | **69.6** | 79.57 |
| FedMVL | 56.21 | 45.88 | 68.05 | 62.13 | 42.28 | 71.14 | 64.90 | 8.71 | 64.90 | 29.71 | 11.20 | 30.36 |
| FedFuzzy | **80.47** | **66.09** | **80.47** | **87.15** | **82.91** | **87.15** | **72.60** | **17.05** | **72.60** | **80.83** | 64.51 | **81.00** |

**Table 1: Clustering performance comparison in terms of ACC(%), NMI(%), and PUR(%) on 3-sources, BBCSport, Sonar and Caltech-5V datasets.**

| Datasets | Yale | | | Vehicle Sensor | | | HAR | | | RGBD | | |
|---|---|---|---|---|---|---|---|---|---|---|---|---|
| Metrics | ACC | NMI | PUR | ACC | NMI | PUR | ACC | NMI | PUR | ACC | NMI | PUR |
| DiMSC | 48.28 | 51.85 | 49.09 | 76.06 | 29.47 | 76.06 | 51.79 | 32.14 | 25.69 | 40.72 | 32.57 | 50.10 |
| MvLRSSC | 45.85 | 50.16 | 46.97 | 56.78 | 6.12 | 56.78 | 49.38 | 53.56 | 53.40 | 43.95 | **37.29** | 43.29 |
| RMSL | 67.27 | **74.02** | 68.48 | 68.07 | 12.34 | 68.07 | 48.64 | 52.99 | 55.38 | 13.80 | 3.06 | 26.43 |
| GMC | 54.55 | 62.44 | 54.55 | 64.68 | 19.55 | 64.68 | 48.04 | **57.40** | 48.60 | 40.23 | 33.06 | 46.51 |
| MvDGNMF | 47.27 | 52.24 | 50.91 | 52.63 | 0.20 | 52.63 | 46.36 | 35.21 | 46.36 | 26.57 | 0.78 | 26.98 |
| UDBGL | 52.73 | 65.94 | 54.55 | 51.69 | 0.08 | 51.69 | 47.78 | 46.20 | 50.45 | 43.89 | 35.96 | **53.55** |
| FastMICE | 66.68 | 67.94 | 67.88 | 51.76 | 0.09 | 51.76 | 56.66 | 50.17 | 56.71 | 41.81 | 32.61 | 49.53 |
| FedMVL | 46.67 | 51.50 | 47.27 | 74.03 | 17.39 | 74.03 | 53.68 | 54.70 | 43.71 | 32.51 | 23.65 | 45.89 |
| FFCMSP | **67.64** | 73.67 | **73.91** | **99.94** | **99.31** | **99.94** | **59.54** | 53.22 | **59.54** | **46.71** | 27.13 | 51.84 |

**Table 2: Clustering performance comparison in terms of ACC(%), NMI(%), and PUR(%) on Yale, Vehicle Sensor, HAR, and RGBD datasets.**

**Datasets:** We evaluate our method on eight public multi-view datasets, Concretely: (1)**3-sources** is a three-view text dataset and the three views are sourced from three reputable news outlets: BBC, Reuters, and The Guardian while their dimensions are 3056, 3631, and 3068. 169 samples are selected to form the datasets. (2)**BBCSport** [16] is composed of 544 sports news articles sourced from the BBC Sport website spanning the years 2004-2005. The dataset is categorized into five distinct topical areas with the following class labels: athletics, cricket, football, rugby, and tennis. It has two views and the dimensions are 3283 and 3183 respectively (3)**Sonar** [37] extracts its multi-view features from the 111 patterns obtained by bouncing sonar signals off a metal cylinder at various angles and under various conditions and 97 patterns obtained from rocks under similar conditions. Then the 60 features are divided into three views equally. (4)**Caltech-5v** [13] is the five-view version of Caltech-7, which has 7 classes and 1400 sample. The dimensions of Caltech-5v are [40, 254, 1984, 512, 1400]. (5)**Yale** is a two-view dataset of 165 facial images of 11 people, and the first view is a $32 \times 32$ image, while the second view is a $64 \times 64$ image. (6)**Vehicle Sensor** [9] is a four-view dataset and the four features are gathered from distributed sensors while their dimensions are 5, 5, 7, and 5. (7)**Human Activity Recognition(HAR)** [36] is a dataset documenting 30 individuals performing six daily activities(walking, walking upstairs, walking downstairs,

sitting, standing, lying down), and the features are gathered from Samsung Galaxy S II on the waist. The four-view dataset consists of 10299 samples. The feature dimensions of the four views are 20, 65, 237, and 59. (8)**SentencesNYU v2(RGB-D)** [38] is about images of indoor scenes and corresponding descriptions. We preprocess the dataset by following [40] and obtained a two-view dataset of 1449 samples divided into 13 classes, and the feature dimensions are 2048, and 300.

**Compared Methods:** We compared our methods with five centralized multi-view clustering methods and two federated multi-view clustering to verify the superiority of our proposed model. Concretely, (1)**DiMSC** [3] boosts the multi-view clustering by mining the complementary information among multi-view features. (2)**MvLRSSC** [2] constructs an affinity matrix shared among all views to learn a joint subspace representation. (3)**RMSL** [26] is composed of Hierarchical Self-Representative Layers and Backward Encoding Networks, which recover the subspace structure of data and explore the complex relationships among different views respectively. (4)**GMC** [42] is a graph-based multi-view clustering method that learns a unified graph from the affinity graph of each view for clustering. (5)**MvDGNMF** [25] is an NMF-based method that can extract more abstract representation by constructing a multi-layer NMF model with graph Laplacian regularizer. (6) **UDBGL** [11]

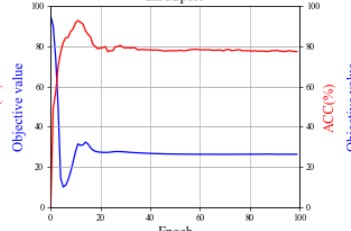
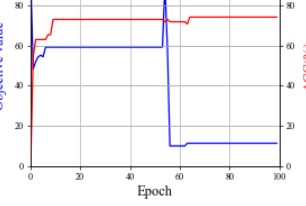
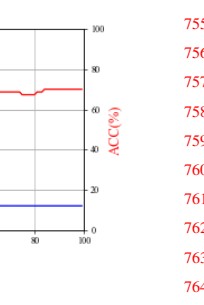
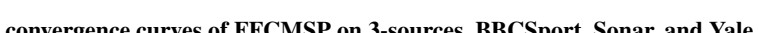

**Figure 2: The convergence curves of FFCMSP on 3-sources, BBCSport, Sonar, and Yale.**

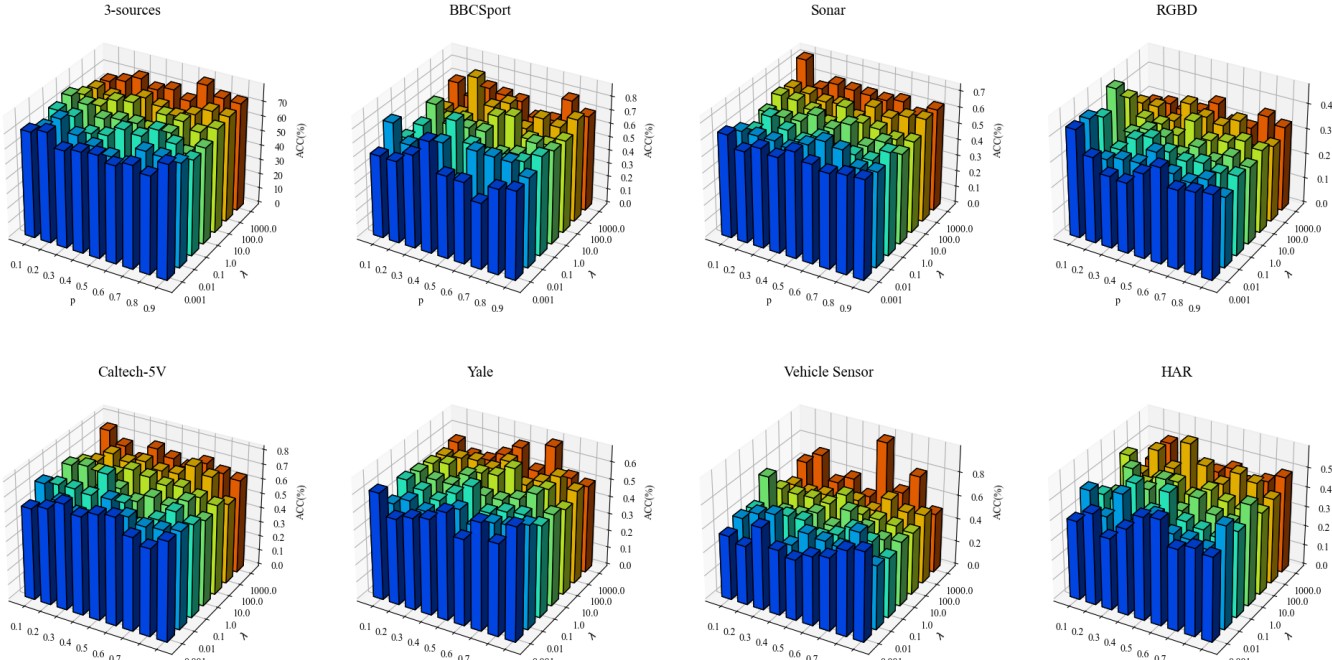

**Figure 3: Parameter sensitivity analysis concerning $p$ and $\lambda$.**

jointly learns both single-view and consensus graphs and combines it with k-means to construct a unified framework. (7) **FastMICE** [18] presents the concept of a random view group to capture the versatile view-wise relationships and design a hybrid early-late fusion strategy. The method has linear time and space complexity. (8)**FedMVL** [19] is a federated multi-view clustering based on NMF and K-means.

## 5.2 Experiment Results and Analysis

**Table.** 1 and **Table.** 2 present the experiment results of the proposed method and eight other multi-view clustering methods on eight real-world datasets. From the results we can observe that on ACC, NMI, and PUR, FFCMSP demonstrates superior performance across almost all datasets, demonstrating its robust clustering capability. Additionally, on the sonar dataset, it outperforms the second-best model by 7.70%, 8.34%, and 7.70%, respectively. Particularly, its performance on the Vehicle Sensor dataset, where its ACC, NMI, and PUR metrics nearly reach 100%, far surpassing other algorithms.

This could be attributed to the proposed FFCMSP effectively capturing the consistency and complementary information between views using tensor Schatten $p$-norm while leveraging FCM to better preserve information between data points. Besides, in the eight chosen compared method, FedMVL is the only federated multi-view clustering method, thus, the experiment results further demonstrate the the superiority of FFCMSP over centralized ones. Overall, these experimental results validate the superiority of FFCMSP in multi-view clustering tasks.

## 5.3 Convergence analysis

We record the values of $\sum_{m=1}^{M} \sum_{i=1}^{N} \sum_{j=1}^{C} u_{ij}^{(m)} ||\mathbf{x}_i^{(m)} - \mathbf{v}_j^{(m)}||_2^2 + \lambda ||\mathcal{U}||_{sp}^p$ and the ACC at each iteration on four datasets. The results are also displayed in **Fig** 2, from which we observed that FFCMSP converges quickly on these four datasets, typically reaching convergence within 20 iterations. Moreover, if the early-stop trick is

| Variants | case 1 | | | case 2 | | | FFCMSP | | |
|---|---|---|---|---|---|---|---|---|---|
| Dataset | ACC | NMI | PUR | ACC | NMI | PUR | ACC | NMI | PUR |
| 3-cources | 61.54 | 34.81 | 61.54 | 69.82 | 53.38 | 69.82 | **80.47** | **66.09** | **80.47** |
| BBCSport | 41.73 | 9.45 | 41.73 | 66.54 | 37.90 | 66.54 | **87.15** | **82.91** | **87.15** |
| Sonar | 54.33 | 0.58 | 54.33 | 66.35 | 7.75 | 66.35 | **72.60** | **17.05** | **72.60** |
| Caltech-5v | 46.36 | 30.07 | 51.71 | 60.07 | 49.78 | 60.07 | **80.83** | **64.51** | **81.00** |
| Yale | 38.79 | 48.83 | 38.79 | 39.39 | 40.14 | 39.39 | **67.64** | **73.67** | **73.91** |
| Vehicle | 52.07 | 16.50 | 52.07 | 74.53 | 20.56 | 74.53 | **99.94** | **99.31** | **99.94** |
| HAR | 52.40 | 59.70 | 52.40 | 52.40 | 35.21 | 54.16 | **59.54** | 53.22 | **59.54** |
| RGBD | 36.71 | 22.54 | 39.82 | 38.16 | 22.84 | 42.93 | **46.71** | **27.13** | **51.84** |

**Table 3: Results of ablation studies.**

employed, the clustering accuracy of FFCMSP would further improve(The experimental results documented in this paper are all based on converged results). We can still observe that neither the convergence curves of the objective function nor the ACC exhibit oscillations, which typically arise from federated aggregation. This indicates the robustness of our model to federated settings.

## 5.4 Parameter Sensitivity Analysis

The objective function **Eq.** (9) has two main parameters: $p$ and $\lambda$. We analyze their impact on the clustering results, which are depicted in **Fig.** 3. Firstly, we analyze the influence of two parameters. $p$ is the most significant parameter in the tensor Scatten-p norm. When $0 \le p \le 1$, the tensor Scatten p-norm has better rank approximation. $\lambda$ is the trade-off parameter to balance the effects of the two items. When $\lambda$ is large, the tensor Schatten $p$-norm dominates, leading the model to explore more spatial structural information between views, while when $\lambda$ is small, the model focuses on local training. From **Fig.** 3, we conclude that the chosen for $p$ is closely related to the dataset, but when $\lambda$ is set to a larger value, the model performs better. Above all, the suggested range of $\lambda$ is [100, 1000].

## 5.5 Ablation Experiments

Our proposed FFCMSP consists of two main modules: The fuzzy C-Means module and the tensor Schatten $p$-norm. As the tensor $\mathcal{U}$ is constructed from the all view-specific membership degree matrix U, thus the tensor Schatten $p$-norm cannot exist independently. Consequently, we conduct the ablation study in the following two cases: (1) FFCMSP w/o tensor Schatten-$p$ regularizer and fuzzy C-Means(**case 1**); (2)FFCMSP w/o tensor Schatten $p$-norm regularizer(**case 2**); It is obvious that **case 1** is equal to multi-view K-means, and the results are shown in **Table.** 3. From the results, we observe that on 3-sources, the ACC, NMI, and PUR of **case 2** outperform that in **case 1** by 7.28%, 18.57%, and 7.28 %, demonstrating that introducing membership degree to clustering surely enhances clustering performance. Moreover, the ACC, NMI, and PUR of FFCMSP on 3-sources also increase by 10.65%, 12.71%, and 10.65% compared with **case 2**, indicating that introducing the tensor Schatten $p$-norm can better explore spatial structural information between views. In summary, the ablation experiments strongly support the significant role of the two key modules in enhancing clustering performance.

## 6 CONCLUSION

In this paper, we proposed a novel fuzzy C-Means and Schatten $p$-norm based federated multi-view clustering method named FFCMSP. This method integrates fuzzy C-Means clustering and Schatten $p$-norm regularizer to enhance the clustering performance of multi-view data in a federated learning setting. Specifically, by employing fuzzy C-Means, we effectively alleviate the information loss caused by K-means, while the Schatten $p$–norm helps to exploit the inter-view complementary information and spatial structure, thereby improving the accuracy and robustness of clustering. Besides, the proposed federated optimization algorithm enables clients to train the global clustering model collaboratively. Furthermore, we evaluated the FFCMSP method on multiple real-world datasets and compared it with several state-of-the-art multi-view clustering methods. The experimental results demonstrate that our method generally achieves superior clustering performance.

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
