# OpenReview forum: "Federated Fuzzy C-means with Schatten-p Norm Minimization"
_acmmm.org/ACMMM/2024/Conference — MM2024 Poster_

### Official Review · Reviewer_YLEt · 2024-05-24

**Rating:** 2
**Confidence:** 3

**Summary:**

This paper proposes a method termed Federated Fuzzy C-Means with Schatten-p Norm Minimization (FFCMSP). The proposed method can tackle two key challenges in federated multi-view clustering: data privacy and the need to leverage both consistency and complementarity in multi-view data. Several experiments are conducted to verify the effectiveness of the proposed method.

**Strengths:**

1. The proposed method improves clustering accuracy.
2. The proposed method can protect data privacy.

**Limitations:**

1. The paper has limited innovation. The proposed method appears to be merely a traditional multi-view clustering. I acknowledge that this method can handle federated learning tasks; however, its main innovation does not lie in this aspect.
2. Can you explain why the objective function is not monotonically decreasing? I suspect there are some errors in the implementation of the algorithm.
3. Some grammar errors or typos need to be corrected. "... sensitivity to outliers and "  ---> “sensitivity to outliers” on Line 89.
4. The writing of the paper needs improvement. The related work section should describe specific methods closely related to the proposed approach, rather than conducting another literature review.

**Suitability:**

2

---

### Official Review · Reviewer_NsWC · 2024-05-24

**Rating:** 5
**Confidence:** 4

**Summary:**

In this paper, authors propose a federated fuzzy C-means for multi-view clustering. Its main contributions lie in that it first introduces a tensorized Schatten p-norm regularization to help the fuzzy C-means to better capture the inter-view spatial features embedded in multi-view data. In this way, the complementary information can be better learned by the clustering model, thus achieving better clustering performance. Furthermore, the authors proposed a federated fuzzy C-means, which well addresses the issue that the multi-view data is held by different parties and supports collaborative training without exchanging the raw data. The proposed method helps to extend the potential applications of multi-view clustering. To illustrate the superiority of the work, the paper compares it with both centralized and federated methods via several experiments.

**Strengths:**

1. The paper is with a clear organization and is relatively easy to follow.
2. The work explores the application of multi-view clustering under the federated scenarios and discusses a practical deployment problem of multi-view clustering. The provided multi-view fuzzy C-means method contributes to solving this problem with fuzzy C-means enhanced by tensorized schatten-p norm as well as carefully designed federated optimization algorithm.
3. The paper is technically sound. Especially, the paper includes a formal derivation indicating how the optimization algorithm is developed.
4. Extensive experiments are conducted.

**Limitations:**

1. There are some typos in this paper. For example, Figure 2 depicts the relationship between ACC/Objective and the number epochs. However, the description on Figure 2 mentions its is with regard to the number of iterations. The authors should current this inconsistency issue; Eq. (8) lacks the parameter to be optimized.
2. The section on related work could be improved by including more references to multi-view learning and federated multi-view clustering, particularly those on Schatten p-norm regularization. This would provide a more comprehensive background and show the broader applicability of the proposed method.
3. The federated workflow should be explained in more detail. It is suggested to integrate this description with Figure 1 to provide a clearer understanding of the workflow and how it operates in a federated learning scenario.

**Suitability:**

3

---

### Official Review · Reviewer_Bp7W · 2024-05-25

**Rating:** 6
**Confidence:** 3

**Summary:**

The paper focuses on two problems in federated multi-view clustering. First, multi-view data might be distributed across different clients, which hinders the training of multi-view clustering model. For another, multi-view data contain both consistent and complementary information. To overcome these two problems, the authors proposed a novel federated multi-view method named Federated Fuzzy C-Means with Schatten-p Norm Minimization. This method adopts fuzzy C-means to alleviate the weakness of K-means. With the tensor schatten p-norm minimization, it can better exploit the inter-view information. A federated optimization algorithm is designed to support the training of the proposed multi-view fuzzy C-means without leaking any information.

**Strengths:**

1.The proposed method can solve the weaknesses of multi-view K-means and can be trained by multiple clients without leaking the private local data.
2.The collaborative optimization algorithm is computationally efficient and does not require much memory.
3.The introduction of the tensorial constraint helps to explore the complementary information of different views.
4.The experimental results illustrate the effectiveness of the proposed method.

**Limitations:**

1.For the tensor schatten-p norm used in the method, the parameter $p$ seems to have an important influence on the final performance. Especially, wheen $p=1$, the tensor Schatten p-norm is tensor nuclear norm, and when $p=0$, it is the tensor rank. However, the influence of $p$ in tensor Schatten p-norm on the model performance is not revealed in the experiment.
2.Figure 1 should be further improved and clarified. The diamond symbol introduced in the title of Figure 1 is not consistent with that in the figure. Besides, some information in the Algorithm 1 is not revealed in this Figure. For example, whether $Q^m$ is transmitted between clients and global server is not well illustrated.
3.For the experiments, the authors only compare the proposed method with one federated method, which may not be enough to evaluate the performance. Maybe more federated methods should be compared. Besides, since the authors argued that the proposed federated multi-view C-means could ensure data privacy, the security issue should also be analyzed or discussed.

**Suitability:**

3

---

### Official Review · Reviewer_569V · 2024-05-25

**Rating:** 3
**Confidence:** 3

**Summary:**

The research addresses the challenges of federated multi-view clustering. A new method called FFCMSP is proposed. This method improves traditional K-means by using fuzzy C-means to better manage uncertainty and reduce information loss. The incorporation of a Schatten 𝑝-norm regularizer helps to leverage inter-view complementary information and the global spatial structure of the data. A federated optimization algorithm is developed to allow clients to collaboratively train the model without data leakage. Experimental results on various datasets demonstrate that FFCMSP outperforms existing multi-view clustering methods in accuracy and robustness.

**Strengths:**

The paper is written in a clear and comprehensible manner, with a logical flow of ideas, and includes necessary theoretical proofs and experimental analyses.

**Limitations:**

1. Reference [17] in the paper also uses the fuzzy C-means method for multi-view clustering. It appears that this paper merely adds a Tensor Schatten $p$-Norm but does not provide a comparative analysis with that method. Additionally, it mentions a comparison with two federated multi-view methods on line 681, but on line 798, it states that only one federated multi-view method was compared.

2. Is it that common federated learning methods cannot be directly applied to multi-view data? Why? And I am interested in how this proposed method compares with other non-federated learning algorithms.

3. The title of the paper does not indicate that this is a federated learning algorithm designed for multi-view data, thus failing to reflect the characteristics of the method.

**Suitability:**

2

---

### Meta-Review · Area_Chair_kXoC · 2024-07-09

**Recommendation:** Accept (Poster)
**Confidence:** 5

**Metareview:**

This paper proposes a method termed Federated Fuzzy C-Means with Schatten-p Norm Minimization (FFCMSP). The proposed method can tackle two key challenges in federated multi-view clustering: data privacy and the need to leverage both consistency and complementarity in multi-view data. This article has received 4 reviews. The 4 reviewers are confident. 2 reviewers reject this paper, and 2 reviewers accept it. I suggest to accept as the positive balance seems better.

---

### Meta-Review · Senior_Area_Chairs · 2024-07-10

**Recommendation:** Accept (Poster)
**Confidence:** 4

**Metareview:**

This paper received mixed ratings initially. After rebuttal, two reviewers tend to accept the paper, one did not submit the final ratings, one gave BR. SAC and AC recommend acceptance of the paper.